# Expression of Vascular Endothelial Growth Factor-C in the Trabecular Meshwork of Patients with Neovascular Glaucoma and Primary Open-Angle Glaucoma

**DOI:** 10.3390/jcm10132977

**Published:** 2021-07-02

**Authors:** Keitaro Hase, Satoru Kase, Atsuhiro Kanda, Yasuhiro Shinmei, Kousuke Noda, Susumu Ishida

**Affiliations:** Laboratory of Ocular Cell Biology and Visual Science, Department of Ophthalmology, Faculty of Medicine and Graduate School of Medicine, Hokkaido University, Sapporo 060-8638, Hokkaido, Japan; k.hase0730@gmail.com (K.H.); kanda@med.hokudai.ac.jp (A.K.); yshinmei@med.hokudai.ac.jp (Y.S.); nodako@med.hokudai.ac.jp (K.N.); ishidasu@med.hokudai.ac.jp (S.I.)

**Keywords:** VEGF-C, VEGFR3, trabecular meshwork, neovascular glaucoma, primary open-angle glaucoma

## Abstract

To investigate the expression of vascular endothelial growth factor (VEGF)-C and vascular endothelial growth factor receptor (VEGFR)3 in the trabecular meshwork (TM) of patients with glaucoma and cultured TM cells. **Methods:** The expressions of VEGF-C in angle tissues collected by trabeculectomy from patients with glaucoma and non-glaucomatous choroidal malignant melanoma were analyzed by immunohistochemistry. Additionally, VEGF-C concentrations were determined in the aqueous humor of patients with glaucoma by ELISA. The expressions of VEGFR3, which is a receptor of VEGF-C in cultured TM cells, were analyzed by Western blot analysis and immunocytochemistry. Cultured TM cells were stimulated by oxidative stress, hypoxia, or high glucose conditions, and VEGF-C concentrations in supernatants and cell lysates were determined by ELISA. **Results:** VEGF-C immunoreactivity was positive in TM tissues of glaucoma patients, but not in those of non-glaucomatous controls. VEGF-C concentrations in the aqueous humor of patients with neovascular glaucoma and primary open-angle glaucoma were lower than those with non-glaucoma patients. VEGFR3 was expressed in cultured TM cells. VEGF-C concentrations in supernatants or cell lysates of TM cells cultured under oxidative stress and hypoxia were significantly elevated compared with those under steady conditions (*p* < 0.05). VEGF-C concentrations in supernatants and cell lysates of TM cells cultured in high glucose were significantly higher than those in low glucose (*p* < 0.01). **Conclusions:** VEGF-C was expressed in TM tissues of patients with glaucoma, which was secreted from cultured TM cells under various pathological conditions. These results suggest that VEGF-C may be involved in the pathology of glaucoma.

## 1. Introduction

Glaucoma is a progressive degenerative optic neuropathy and is a leading cause of irreversible blindness in the world [1]. One of the mechanisms underlying pathogenesis of glaucoma is an increased intraocular pressure caused by disruption of aqueous outflow in the trabecular meshwork (TM) [2]. Glaucoma is commonly divided into two major subtypes: open angle and angle closure [3]. Neovascular glaucoma (NVG) is a form of secondary angle-closure glaucoma characterized by neovascularization of the angle, which causes occlusion of the TM [4]. NVG is a refractory disease and often leads to blindness. Proliferative diabetic retinopathy (PDR) and retinal vein occlusion with angiogenesis are important as causative diseases of NVG [5]. It has been shown that vascular endothelial growth factor (VEGF)-A, which is one of the molecules leading to angiogenesis, is mainly involved in the onset of NVG [6]. Actually, it has been reported that the mean concentration of VEGF-A in aqueous humor of patients with NVG was significantly higher than that in patients with primary open-angle glaucoma (POAG) and cataract [7]. The treatment outcomes are improving with intravitreal injections of anti-VEGF-A antibody [8]; however, there are still many cases showing resistance to anti-VEGF-A therapy [9]. POAG is a chronic optic neuropathy and the most common form of glaucoma [10]. Though new treatments for glaucoma such as eyedrops, laser therapy, and surgery have been developed to slow visual field loss by lowering intraocular pressure, many patients still lose their eyesight due to glaucoma [11,12].

VEGF-C has been shown to provoke lymphangiogenesis and contribute to the tumor growth [13,14,15]. Previously, it has been reported that VEGF-C was elevated in the vitreous of NVG patients secondary to proliferative diabetic retinopathy (PDR) [16]. However, there is no report on the analysis of VEGF-C expression in the TM of patients with glaucoma such as NVG and POAG. Moreover, the involvement of VEGF-C in the pathogenesis of glaucoma has not been elucidated. Therefore, clarifying the mechanism of VEGF-C expression in the TM of patients with glaucoma may contribute to discovery of a new therapeutic target for refractory glaucoma. The aim of this study was to investigate VEGF-C expression in the TM of patients with glaucoma.

## 2. Materials and Methods

### 2.1. Human Surgical Sample

Residual angle tissues were collected during trabeculectomy from patients with NVG due to PDR (68-year-old man) and POAG (58-year-old man). Eyeball tissues were gained by enucleation from a patient with choroidal malignant melanoma (72-year-old man) at the Hokkaido University Hospital. The excised tissue was fixed with 4% paraformaldehyde, and then a paraffin-embedded specimen was prepared. All clinical specimens used in this study were collected from patients with written informed consent, and the use of pathological specimens was approved by the institutional review board (IRB No. 016-0128) of Hokkaido University, Japan.

The aqueous humor of glaucoma patient (NVG: *n* = 10 (4 PDR, 3 radiation retinopathy, 2 central retinal vein occlusion, 1 ocular ischemic syndrome), 64.3 ± 8.0 years, POAG: *n* = 6, 59.3 ± 9.5 years) and non-glaucoma patient (cataract): *n* = 9, 72.0 ± 7.2 years)) was collected during surgery. The information of patients enrolled in this study are shown in Table 1.

### 2.2. Immunofluorescence Microscopy

Paraffin-embedded specimens were deparaffinized with xylene and ethanol, and then antigen-activated with a 10 mM citrate buffer (pH 6.0) in a microwave oven. Next, 1% bovine serum albumin (BSA) (Wako, Osaka, Japan) diluted with PBS was added dropwise to prevent nonspecific protein binding, and the mixture was placed in a humid box for 30 min. Next, goat anti-VEGF-C (AF752) (1:50) (R&D Systems, Minneapolis, MN, USA) diluted with PBS was added dropwise as a primary antibody, and the mixture was reacted overnight in a humid box at 4 °C. Then, a secondary antibody reaction was performed for 30 min at room temperature using Alexa Fluor 488 or 546 (1:400) (Life Technologies, Carlsbad, CA, USA). For immunocytochemistry of trabecular meshwork cells, cells were seeded into a six-well plate with cover glass and fixed with 4% paraformaldehyde for 15 min and permeabilized by 0.1% Triton X-100 for 10 min. Cells were incubated in 5% BSA/PBS for 30 min and then incubated with a primary goat antibody against VEGFR3 (1:50) (AF349) (R&D systems) at 4 °C overnight prior to the exposure to Alexa Fluor 546 anti-goat IgG (1:400) for 1 h at room temperature. Nuclear staining was performed with 4′,6-diamino-2-phenylindole (DAPI) (1:500) (Lonza, Basel, Switzerland).

The observation of the section was performed with a fluorescence microscope Keyence BZ-9000 series (BIOREVO, Keyence, Osaka, Japan).

### 2.3. Enzyme-Linked Immunosorbent Assays (ELISA)

The protein concentrations of VEGF-C were determined with human VEGF-C ELISA kit (R&D Systems) per the manufacturer’s instruction. The optical density was determined using a micro-plate reader (Sunrise, TECAN, Männedorf, Switzerland).

### 2.4. Cell Lines

Immortalized TM cells GTM3 (Novartis, Basel, Switzerland) established from POAG patient eyes were incubated in D-MEM (Dulbecco’s Modified Eagle’s Medium) (high glucose) (Wako, Osaka, Japan) supplemented with 10% fetal bovine serum (FCS) (Thermo Fisher Scientific, MA, USA) [17].

### 2.5. Oxidative Stress Experiments

TM cells (4 × 10^4^/well) were cultured on three 6-well plates under normal conditions. Oxidative stress load is 30% H_2_O_2_ (Wako, Osaka, Japan) diluted to make 1, 10, 100 μM concentrations for 24 h. After the supernatants in all wells were aspirated, one plate was cultured under 0 μM H_2_O_2_ media and other plates were cultured under 1, 10, 100 μM H_2_O_2_ media for 24 h. Then, the culture supernatants and the cell lysates were collected, and VEGF-C concentrations were determined by ELISA.

### 2.6. Hypoxic Condition

TM cells (4 × 10^4^/well) were cultured using two 6-well plates in an O_2_/CO_2_ incubator (Wakenyaku, Kyoto, Japan). After 24 h, one plate was cultured under normoxia (20% oxygen concentration) and the other plate was cultured under low oxygen conditions (1% oxygen concentration) for 24 h. Then, VEGF-C concentrations in the culture supernatants and the cell lysates were determined by ELISA.

### 2.7. Hyperglycemic Stimulation

TM cells (4 × 10^4^/well) were cultured using two 6-well plates. After 24 h, the supernatants in all wells were aspirated, D-MEM (high glucose: D-Glucose 4500 mg/mL) was added to wells in one plate (hyperglycemic condition), and D-MEM (low glucose: D-glucose 1000 mg/mL) (Wako, Osaka, Japan) was added to wells in the other plate (hypoglycemic condition). After 24 h, the culture supernatants and the cell lysates were collected, and VEGF-C concentrations were determined by ELISA.

### 2.8. Western Blot Analysis

Collected cells were lysed in SDS buffer and sonicated on ice. After quantifying protein concentrations using BCA reagent (Thermo Fisher Scientific, Waltham, MA, USA), proteins were resolved by SDS-PAGE and transferred to PVDF membrane by electroblot analysis. Membranes were blocked in Tris-buffered saline containing 5% skim milk and 0.05% Tween-20 and incubated with the following primary antibodies: goat anti-VEGFR3 (1:1000) (AF349) (R&D systems) and rabbit anti-β actin (1:8000) (Medical and Biological Laboratories, Nagoya, Aichi, Japan). Horseradish peroxidase conjugated anti-goat, anti-rabbit IgG (1:4000) were used as secondary antibodies for chemiluminescence detection. Signals were obtained by enhanced chemiluminescence (PerkinElmer, Waltham, MA, USA). The bands were analyzed by densitometry using ImageJ software (National Institutes of Health (NIH), Bethesda, MD, USA).

### 2.9. Evaluation of Apoptosis on TM Cells under Oxidative Stress

TM cells (2 × 10^4^/well) were cultured on 96-well microtiter plates under normal conditions for 24 h. Then, media were replaced with H_2_O_2_ and cultured for 12 h. After that, apoptosis was evaluated by measuring the caspase 3/7 activity on TM cells using Caspase-Glo^®^ 3/7 assay kit (Promega, Madison, Wisconsin, USA). Caspase-Glo^®^ 3/7 assay was performed according to manufacturer’s protocol and 50 μL of the reagent was added per well and incubated for 1 h at room temperature in the dark. Following the incubation, the luminescence was measured on a microplate reader.

### 2.10. Statistical Analysis

All results are shown as the mean ± standard deviation. The Mann–Whitney *U* test was used for comparison between the two groups; control vs. H_2_O_2_, normoxia vs. hypoxia, and high glucose vs. low glucose-treated groups. Tukey–Kramer test was used for comparing the three or more groups among glaucoma and cataract patients as well as H_2_O_2_ additions. Values of *p* < 0.05 were considered significant.

## 3. Results

### 3.1. Immunoreactivity for VEGF-C in Various Angle Tissues with/without Glaucoma

VEGF-C immunoreactivity was found in the TM, Schlemm’s canal endothelial cells and collecting ducts from the patient with NVG secondary to PDR (Figure 1A–C) and with POAG (Figure 1D–F). As a negative control, the tissue of a non-glaucoma patient with choroidal malignant melanoma was used. VEGF-C immunoreactivity was not observed in the TM or Schlemm’s canal endothelial cells (Figure 1G–I).

### 3.2. VEGF-C Concentrations in the Aqueous Humor of Patients with/without Glaucoma

The mean VEGF-C concentrations in the aqueous humor of NVG patients (345 ± 31 pg/mL) were significantly lower than those of non-glaucoma such as cataract patients (753 ± 154 pg/mL) (*p* < 0.05). The mean aqueous VEGF-C concentrations in POAG patients (355 ± 43 pg/mL) were lower than cataract patients (*p* = 0.05) and almost the same as NVG patients (Figure 2).

### 3.3. VEGF-C Concentrations in TM Cells Followed by Various Stimuli

VEGF-C concentrations in supernatants of TM cells which had been treated with 100 μM H_2_O_2_ (*n* = 4) but not 1, 10 μM H_2_O_2_ (*n* = 4 in each group) were significantly higher than those with 0 μM H_2_O_2_ (*n* = 6) (*p* < 0.05) (Figure 3A)_._ VEGF-C concentrations in cell lysates of TM cells which had been treated with 1, 10, 100 μM (*n* = 4 in each group) were not significantly different from those with 0 μM H_2_O_2_ (*n* = 6) (*p* > 0.05) (Figure 3B).

VEGF-C concentrations in both supernatants and cell lysates of TM cells which had been cultured in low oxygen (1% oxygen concentration) were significantly higher than those in stationary oxygen (20% oxygen concentration) (supernatants: *n* = 6 in each group, *p* < 0.05; cell lysates: *n* = 6 in each group, *p* < 0.01) (Figure 4A,B).

VEGF-C concentrations in both supernatants and cell lysates of TM cells which had been cultured in high glucose media were significantly higher than those in low glucose media (*n* = 6 in each group, *p* < 0.01) (Figure 5A,B).

### 3.4. VEGFR3 Protein Expression in TM Cells

Western blot analysis was further performed using the cell lysates cultured in low glucose and high glucose culture media (*n* = 3 in each group). Three bands considered to be VEGFR3 were observed in both groups (Figure 6A, red arrowheads). However, there was no significance in band intensity between low glucose media and high glucose media (Figure 6B) (*p* > 0.05). In immunocytochemistry, VEGFR3 immunoreactivity was detected in cultured TM cells (Figure 6C, red signals).

### 3.5. Apoptosis on TM Cells under Oxidative Stress

Caspase 3/7 activation, which is a marker of apoptosis, in TM cells treated with 500 μM H_2_O_2_ was significantly increased compared to control (0 μM H_2_O_2_) though the activation was not observed in TM cells treated with less than 400 μM H_2_O_2_ (Figure 7). Since this result showed that apoptosis on TM cells occurred when they were treated with 500 μM H_2_O_2,_ stimulation with 500 μM was considered an optimal H_2_O_2_ concentration to induce oxidative stress burden. Next, in order to investigate whether VEGF-C suppresses apoptosis on TM cells, TM cells were cultured with 500 μM H_2_O_2_ for 12 h and then treated with recombinant human VEGF-C protein for 24 h. However, addition of recombinant human VEGF-C did not change the level of caspase 3/7 activation in TM cells treated with H_2_O_2_ (data not shown).

## 4. Discussion

This study for the first time demonstrated VEGF-C immunoreactivity in the angle tissues including TM, and Schlemm’s canal endothelial cells of patients with NVG and POAG. In glaucoma, one of the factors that increase intraocular pressure is the resistance to aqueous outflow through the TM and Schlemm’s canal in the angle tissues [18,19]. The lymphangiogenic growth factor VEGF-C and its receptor, VEGFR3, are involved in Schlemm’s canal development and enlargement, and an injection of recombinant VEGF-C protein into the anterior chamber was associated with a sustained decrease in intraocular pressure in adult mice [20]. In this study, we investigated the function of VEGF-C which was expressed in the angle tissues, especially the TM of patients with NVG and POAG. There are various causes of glaucoma in addition to increased intraocular pressure, e.g., oxidative stress and hypoxia [21]. Therefore, we initially focused on the effect of oxidative stress and hypoxia on VEGF-C in cultured TM cells.

Regarding the relationship between oxidative stress and VEGF-C, we showed elevated VEGF-C concentrations in supernatants of cultured TM cells treated with 100 μM H_2_O_2_ as oxidative stress. VEGF-C/VEGFR3 and its downstream signaling have been reported to promote cell survival under oxidative stress [22]. Based on these things, VEGF-C is considered to be secreted from TM cells to protect against oxidative stress.

Similarly, regarding the relationship between hypoxia and VEGF-C, we found that VEGF-C concentrations in supernatants and cell lysates of cultured TM cells under hypoxia were significantly elevated compared to those under normoxia. Consistent with our result, hypoxia has been reported to induce VEGF-C expression in tumor cells [23,24]. VEGF-C may be also secreted from TM cells in order to protect against hypoxia caused by glaucoma.

Furthermore, in order to investigate the relationship between VEGF-C and PDR, which is one of the major causative diseases of NVG, TM cells were cultured under high glucose media because the glucose levels in the aqueous humor of diabetic patients have been known to be significantly higher than those of non-diabetic subjects [25]. Therefore, the concentrations of VEGF-C in culture supernatants and cell lysates were examined. As a result, we showed that TM cells stimulated with high glucose produced more VEGF-C than with low glucose.

According to these results, since VEGF-C was released from TM cells under various pathological stimuli, it was expected that VEGF-C concentrations in the aqueous humor of patients with NVG and POAG might be higher than those with non-glaucoma. However, VEGF-C concentrations in the aqueous humor of patients with NVG were significantly lower than those with non-glaucoma patients. The previous report has shown that a high glucose level in the aqueous humor of patients with diabetes may increase fibronectin synthesis and accumulation of the fibronectin in the TM, and accelerate the depletion of TM cells [26]. Taken together, the decrease of VEGF-C concentrations in the aqueous humor of patients with NVG in our study might be explained by the following two reasons: 1) direct use of VEGF-C protein in VEGFR3-expressing TM cells to protect TM cells themselves from the various stimuli through VEGF-C/VEGFR3-signaling pathway, and 2) eventual loss of TM cells producing VEGF-C protein caused by long-term various pathological stress.

In addition, the VEGF-C-VEGFR3 axis is known to be the pathway that suppresses cell apoptosis [27,28]. It has been demonstrated that apoptosis occurred in the TM of patients with glaucoma [29]. Therefore, there is a possibility that VEGF-C regulates the progression of glaucoma by preventing apoptosis although our study could not prove the efficacy of suppression of TM cell apoptosis by VEGF-C addition. If the survival or apoptosis of TM cells after addition of VEGF-C can be analyzed, it may contribute to VEGF-C targeted therapy. There is a report that VEGF-C knockdown decreased expression levels of VEGF-A on retinal pigment epithelial cells [30]. VEGF-C targeted therapy may be able to assist anti-VEGF-A therapy though further examination will be needed. In this study, there is a limitation that the number of samples including excised tissues and aqueous humors of patients is small.

## 5. Conclusions

VEGF-C was expressed in TM tissues of patients with glaucoma, which was secreted from cultured TM cells under various pathological conditions. These results suggest that VEGF-C may be involved in the pathology of glaucoma.

## Figures and Tables

**Figure 1 jcm-10-02977-f001:**
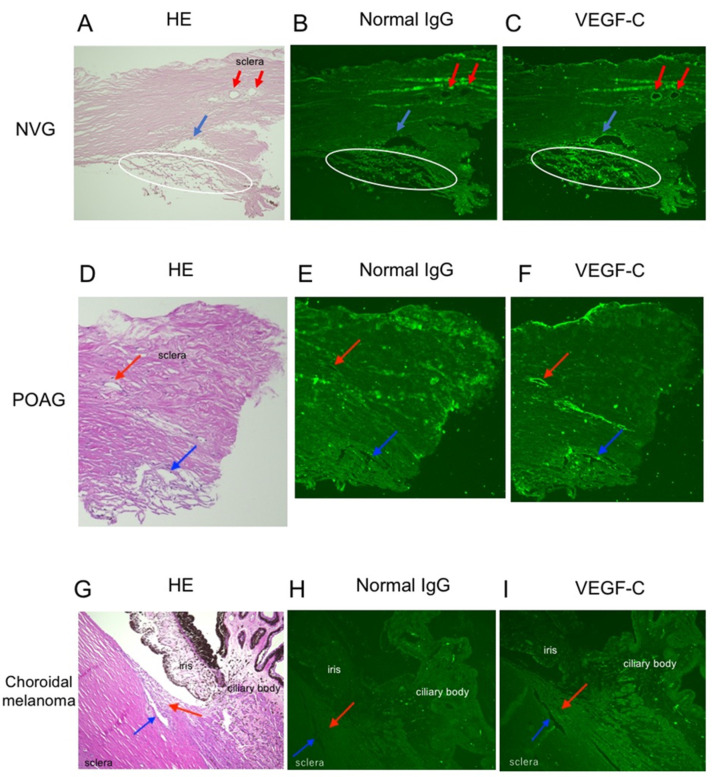
Localization of vascular endothelial growth factor (VEGF)-C in the angle tissues of patients with neovascular glaucoma, primary open-angle glaucoma and non-glaucoma. (**A**–**C**) The angle tissue of a 68-year-old man with neovascular glaucoma secondary to proliferative diabetic retinopathy. (**A**) HE staining. (**B**) Negative control with normal IgG antibody for immunohistochemistry. (**C**) Immunohistochemistry for VEGF-C. Immunoreactivity for VEGF-C proteins was found in the trabecular meshwork, Schlemm’s canal endothelial cells and collecting ducts. White circle indicates the trabecular meshwork. Blue arrow indicates Schlemm’s canal. Red arrow indicates the collecting duct. (**D**–**F**) The angle tissue of a 58-year-old man with primary open-angle glaucoma. (**D**) HE staining. (**E**) Negative control with normal IgG antibody for immunohistochemistry. (**F**) Immunohistochemistry for VEGF-C. Blue arrow indicates Schlemm’s canal. Red arrow indicates the collecting duct. (**G**–**I**) The angle tissue of a 72-year-old man with choroidal malignant melanoma as a control. (**G**) HE staining. (**H**) Negative control with normal IgG antibody for immunohistochemistry. (**I**) Immunohistochemistry for VEGF-C. Blue arrow indicates Schlemm’s canal. Red arrow indicates the trabecular meshwork. A white circle indicates the trabecular meshwork. A blue and a red arrow (**A**–**F**) indicates Schlemm’s canal and the collecting duct, respectively. A red arrow (**G**–**I**) indicates the trabecular meshwork.

**Figure 2 jcm-10-02977-f002:**
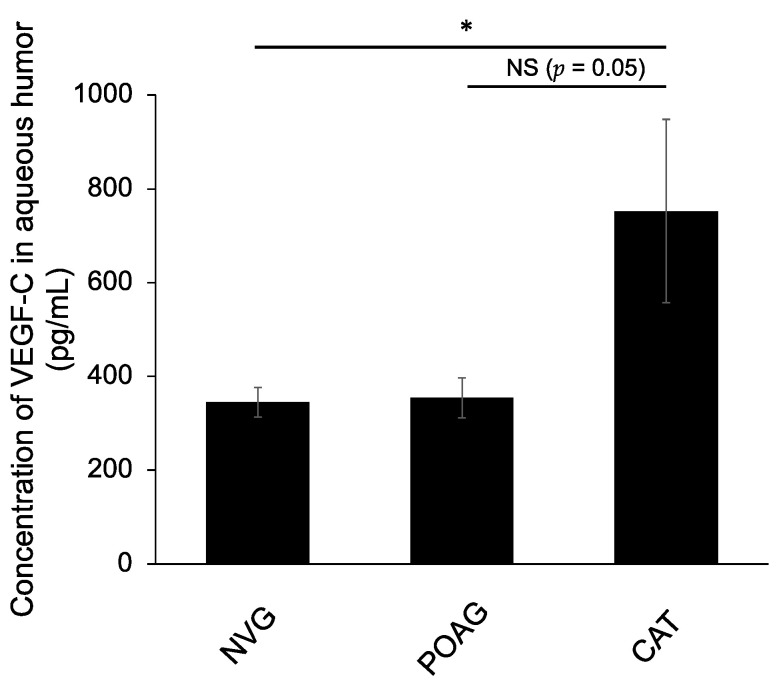
VEGF-C concentrations in the aqueous humor of patients with neovascular glaucoma (NVG), primary open-angle glaucoma (POAG) and non-glaucoma. VEGF-C concentrations of the aqueous humor were measured in patients with NVG (*n* = 10), POAG (*n* = 6) and cataract (*n* = 9). The mean VEGF-C concentrations in the aqueous humor of NVG patients were significantly lower than those of cataract patients (* *p* < 0.05). The mean aqueous VEGF-C concentrations in POAG patients were almost the same as those in NVG patients.

**Figure 3 jcm-10-02977-f003:**
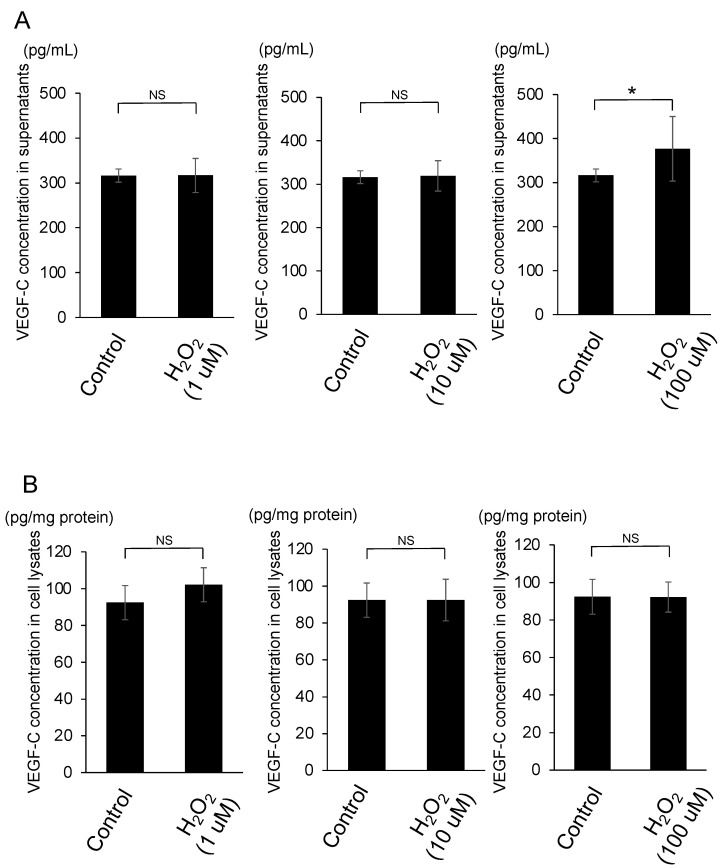
Effect of oxidative stress on VEGF-C concentrations in trabecular meshwork cells. Trabecular meshwork cells were cultured in media with H_2_O_2_ (0, 1, 10, 100 μM) for 24 h. Additionally, VEGF-C concentrations in the supernatants (**A**) and the cell lysates (**B**) were detected using ELISA. VEGF-C concentrations in the supernatants with 100 μM H_2_O_2_ (*n* = 4) but not 1, 10 μM H_2_O_2_ (*n* = 4 in each group) were significantly elevated compared with control (0 μM H_2_O_2_) (*n* = 6) (* *p* < 0.05). VEGF-C concentrations in the cell lysates with all concentrations of H_2_O_2_ (1, 10, 100 μM) (*n* = 4 in each group) were not significantly different from control (0 μM H_2_O_2_) (*n* = 6) (*p* > 0.05).

**Figure 4 jcm-10-02977-f004:**
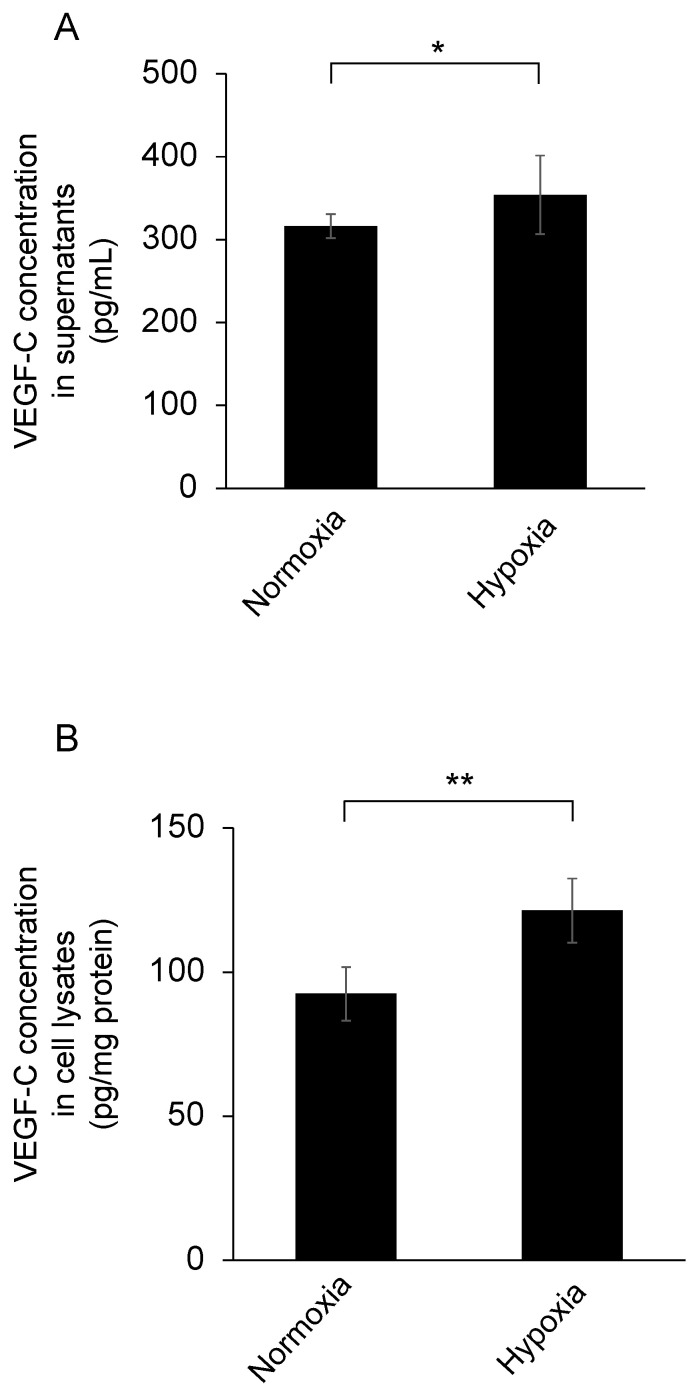
Effect of hypoxia on VEGF-C concentrations in trabecular meshwork cells. Trabecular meshwork cells were cultured under normoxic or hypoxic conditions for 24 h. Additionally, VEGF-C concentrations in the supernatants (**A**) and the cell lysates (**B**) were detected using ELISA. VEGF-C concentrations in trabecular meshwork cells under hypoxic condition were significantly elevated compared with those in normoxic condition in both the supernatants and the cell lysates (*n* = 6 in each group, * *p* < 0.05, ** *p* < 0.01).

**Figure 5 jcm-10-02977-f005:**
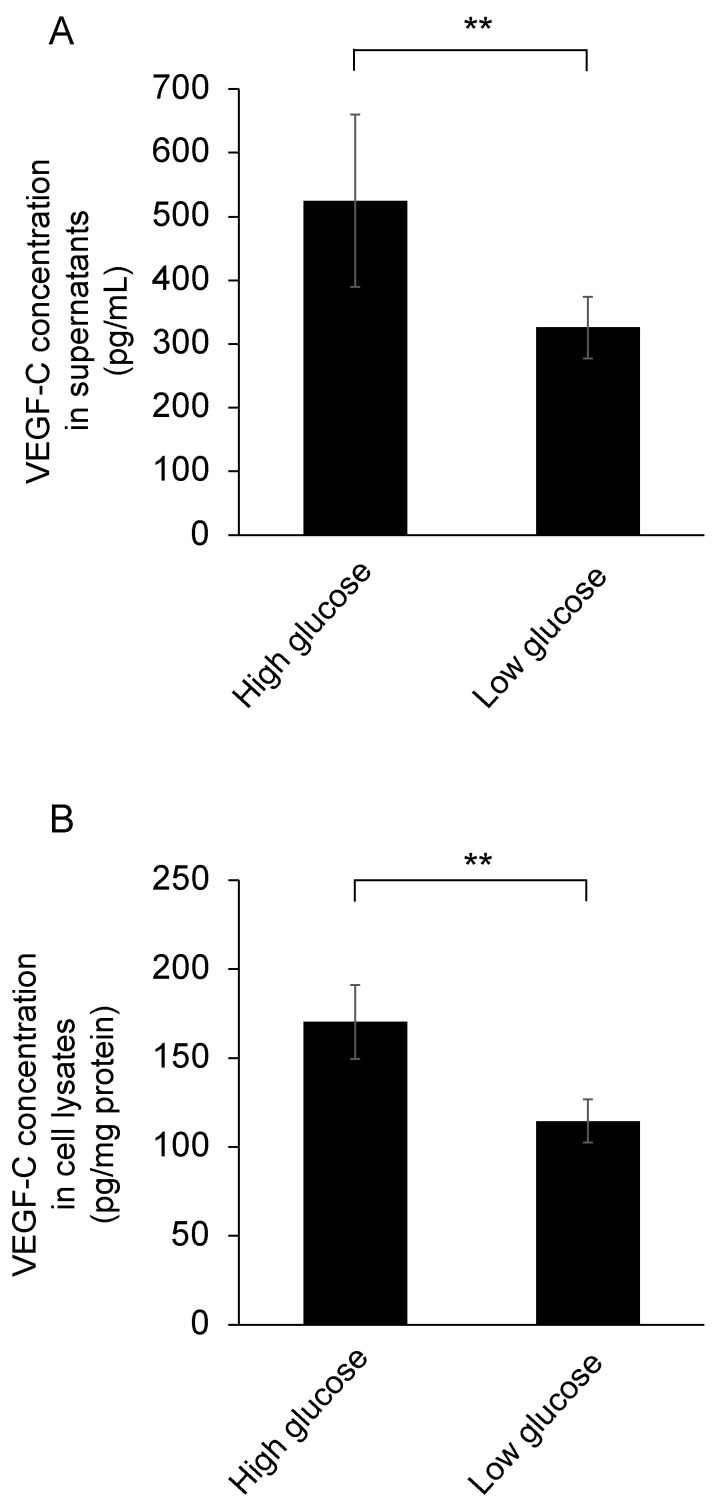
Effect of high glucose on VEGF-C concentrations in trabecular meshwork cells. Trabecular meshwork cells were cultured in high- or low-glucose media for 24 h. Additionally, VEGF-C concentrations in the supernatants (**A**) and the cell lysates (**B**) were detected using ELISA. VEGF-C concentrations in trabecular meshwork cells cultured in high glucose were significantly elevated compared with those in low glucose in both the supernatants and the cell lysates (*n* = 6 in each group, ** *p* < 0.01).

**Figure 6 jcm-10-02977-f006:**
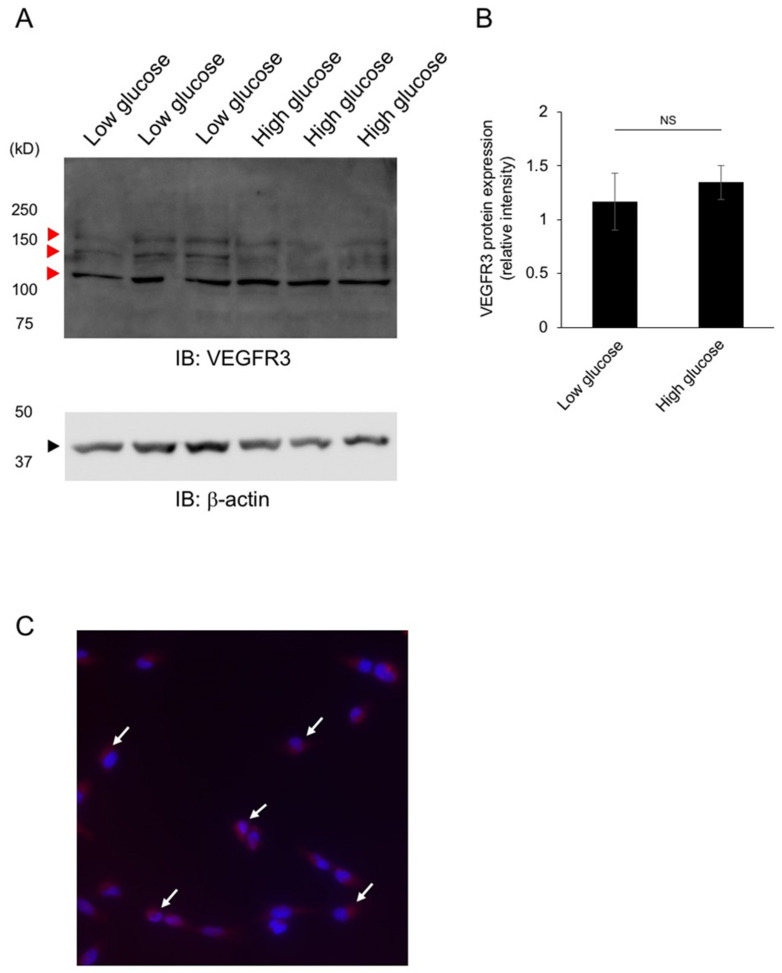
Western blot analysis and immunocytochemistry of trabecular meshwork cells for vascular endothelial growth factor receptor (VEGFR)3. VEGFR3 (red arrowhead) and β-actin (black arrowhead) were detected in the cell lysates of trabecular meshwork cells cultured in low and high glucose media (**A**). Densitometry values of VEGFR3 normalized to β-actin (*n* = 3 in each group) (**B**). There was no significance in band intensity between low glucose media and high glucose media (*p* > 0.05). Immunocytochemistry for VEGFR3 in trabecular meshwork cells. Cytoplasmic immunoreactivity for VEGFR3 (red, white arrows) was noted with DAPI nuclear staining (blue) (**C**).

**Figure 7 jcm-10-02977-f007:**
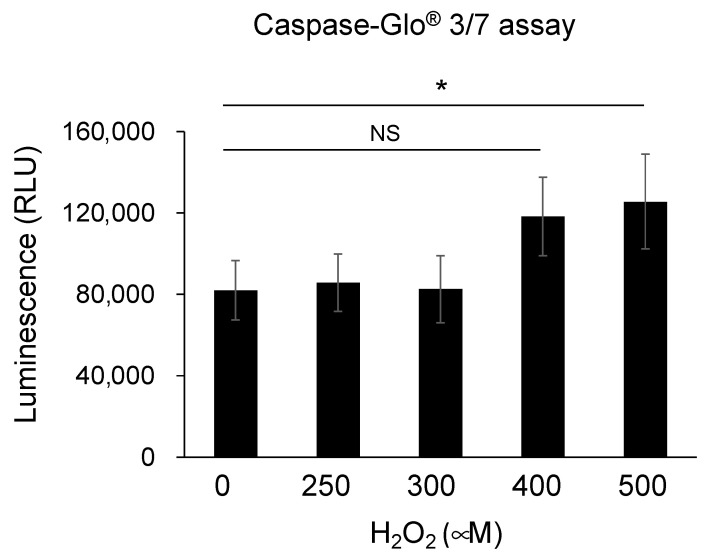
Apoptosis on trabecular meshwork cells by oxidative stress. Caspase 3/7 activation, which is a marker of apoptosis, in trabecular meshwork cells treated with various concentrations of H_2_O_2_ (0, 250, 300, 400, 500 μM) was determined by Caspase-Glo^®^ 3/7 assay. Caspase 3/7 activation on trabecular meshwork cells treated with 500 μM H_2_O_2_ was significantly increased compared to control (0 μM H_2_O_2_) (*n* = 4 in each group, * *p* < 0.05). RLU, relative light units.

**Table 1 jcm-10-02977-t001:** Patients’ demographics for vascular endothelial growth factor (VEGF)-C concentrations in the aqueous humor.

NVG				POAG			CAT		
No.	Gender	Age (Years)	Primary Disease	No.	Gender	Age (Years)	No.	Gender	Age (Years)
1	M	52	PDR	1	M	60	1	M	70
2	F	66	RR	2	M	64	2	M	81
3	M	66	RR	3	F	57	3	F	80
4	M	65	PDR	4	M	50	4	F	62
5	M	72	OIS	5	F	75	5	F	72
6	M	74	CRVO	6	F	50	6	F	78
7	M	61	CRVO				7	F	64
8	M	74	RR				8	F	65
9	M	61	PDR				9	M	76
10	M	52	PDR						

NVG, neovascular glaucoma; POAG, primary open-angle glaucoma; CAT, cataract; M, male; F, female; PDR, proliferative diabetic retinopathy; RR, radiation retinopathy; OIS, ocular ischemic syndrome; CRVO, central retinal vein occlusion.

## Data Availability

Not applicable.

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
