# Peer review of "Expression of Vascular Endothelial Growth Factor-C in the Trabecular Meshwork of Patients with Neovascular Glaucoma and Primary Open-Angle Glaucoma"

_jcm, 2021, doi:10.3390/jcm10132977_

Round 1

Reviewer 1 Report

The objective of the present study is to investigate the expression of vascular endothelial growth factor (VEGF)-C and vascular endothelial growth factor receptor (VEGFR)3 in the trabecular meshwork (TM) of patients with glaucoma and cultured TM cells.

Prior to any consideration for publication, this study should address the following:

Comments of major importance:

  • The study does not include any power analysis that could determine if the sample size is adequate. 

Comments of minor importance:

  • The Table mentioned in line 72 is missing from the pdf text, so it could not be evaluated.
  • The statistical program for the data analysis should be mentioned in the "statistical analysis" subsection. Additionally, you should mention which test was applied for the evaluation of data distribution.
  • The following historical reference could be added: "Tripathi RC, Li J, Tripathi BJ, Chalam KV, Adamis AP. Increased level of vascular endothelial growth factor in aqueous humor of patients with neovascular glaucoma. Ophthalmology. 1998 Feb;105(2):232-7. doi: 10.1016/s0161-6420(98)92782-8."

Author Response

Dear, Aleksandar Jovanovic
JCM Editorial Office

Expression of vascular endothelial growth factor-C in the trabecular meshwork of patients with neovascular glaucoma and primary open-angle glaucoma

We sincerely appreciate for your favorable review of our manuscript and guidance for the revision. Based on suggestions raised by the reviewers, we have carefully revised the manuscript. Newly modified sentences and phrases are shown in red on the next page. Please find below our point-by-point responses to each of the reviewers’ comments reproduced in blue italics, which follows on the next page.

We believe that the reviewers’ thoughtful suggestions have greatly improved the overall quality of our manuscript. We would be grateful if the revised manuscript is considered acceptable for publication in JCM.

Sincerely yours,

Satoru Kase, M.D., Ph.D.

Assistant Professor

Department of Ophthalmology,

Faculty of Medicine and Graduate School of Medicine, Hokkaido University

N-15 W-7, Kita-ku, Sapporo 060-8638, Japan

Tel.: 011-706-5944

Fax: 011-706-5948

Reviewer 1

Comments and Suggestions for Authors

The objective of the present study is to investigate the expression of vascular endothelial growth factor (VEGF)-C and vascular endothelial growth factor receptor (VEGFR)3 in the trabecular meshwork (TM) of patients with glaucoma and cultured TM cells.

Prior to any consideration for publication, this study should address the following:

Comments of major importance:

The study does not include any power analysis that could determine if the sample size is adequate. 

Response: Thank you for your comment. Although the sample size in human materials is not large, there was statistically significant difference between glaucoma and cataract patients. We further did power analysis using the software (Power and Sample Size Calculation 3.1R). Actually, we found that the power was 1.0 in case of P=0.05; therefore, we think that our statistical analyses do not have serious problems. Anyway, as the reviewer suggests, we think that in this study, there is a limitation that the number of samples of patients is small. We have added the following sentence in the Discussion section as a limitation.

Lines 357

In this study, there is a limitation that the number of samples including excised tissues and aqueous humors of patients is small.

Comments of minor importance:

The Table mentioned in line 72 is missing from the pdf text, so it could not be evaluated.

Response: We are sorry for it. We will certainly upload the Table with the revised text in resubmission.

The statistical program for the data analysis should be mentioned in the "statistical analysis" subsection. Additionally, you should mention which test was applied for the evaluation of data distribution.

Response: Thank you for your advice. We described the statistical program for the data analysis in detail. We have added the following sentences in "statistical analysis" section.

Lines 178

The Mann-Whitney U test was used for comparison between the two groups; control vs H2O2, normoxia vs hypoxia, and high glucose vs low glucose-treated groups. Tukey-Kramer test was used for comparing the three or more groups among glaucoma and cataract patients as well as H2O2 additions.

The following historical reference could be added: "Tripathi RC, Li J, Tripathi BJ, Chalam KV, Adamis AP. Increased level of vascular endothelial growth factor in aqueous humor of patients with neovascular glaucoma. Ophthalmology. 1998 Feb;105(2):232-7. doi: 10.1016/s0161-6420(98)92782-8."

Response: Thank you for your advice. We have added the following sentence and the above paper in the reference.

Lines 68

Actually, it has been reported that the mean concentration of VEGF-A in aqueous humor of patients with NVG was significantly higher than that in patients with primary open-angle glaucoma (POAG) and cataract [7].

Reviewer 2 Report

Dear Editor,

Thank you to provide me the opportunity to contribute to your journal. The authors submitted a manuscript about a very interesting topic such as the evaluation of the presence of VEGF receptors in glaucoma eyes.

The study is innovative and well conducted, minor changes are required before considering it suitable for publication.

Lines 33-48: too many sentences without any reference. Same for lines 260-261.

Figure 1 legend: please specify the meaning of red and blue arrows.

Author Response

Dear, Aleksandar Jovanovic
JCM Editorial Office

Expression of vascular endothelial growth factor-C in the trabecular meshwork of patients with neovascular glaucoma and primary open-angle glaucoma

We sincerely appreciate for your favorable review of our manuscript and guidance for the revision. Based on suggestions raised by the reviewers, we have carefully revised the manuscript. Newly modified sentences and phrases are shown in red on the next page. Please find below our point-by-point responses to each of the reviewers’ comments reproduced in blue italics, which follows on the next page.

We believe that the reviewers’ thoughtful suggestions have greatly improved the overall quality of our manuscript. We would be grateful if the revised manuscript is considered acceptable for publication in JCM.

Sincerely yours,

Satoru Kase, M.D., Ph.D.

Assistant Professor

Department of Ophthalmology,

Faculty of Medicine and Graduate School of Medicine, Hokkaido University

N-15 W-7, Kita-ku, Sapporo 060-8638, Japan

Tel.: 011-706-5944

Fax: 011-706-5948

Reviewer 2

Dear Editor,

Thank you to provide me the opportunity to contribute to your journal. The authors submitted a manuscript about a very interesting topic such as the evaluation of the presence of VEGF receptors in glaucoma eyes.

The study is innovative and well conducted, minor changes are required before considering it suitable for publication.

Lines 33-48: too many sentences without any reference. Same for lines 260-261.

Response: Thank you for the comments. The revised version appropriately included the references in introduction and discussion sections as follows:

In introduction, the following references were added (Lines 61-75).

  1. Hann, C.R.; Fautsch, M.P. Recent Developments in Understanding the Role of Aqueous Humor Outflow in Normal and Primary Open Angle Glaucoma. Curr Ophthalmol Rep. 2015, 3, 67-73.
  2. Mantravadi, A.V.; Vadhar, N. Glaucoma. Prim Care. 2015, 42, 437-449.
  3. Shazly, T.A.; Latina, M.A. Neovascular glaucoma: etiology, diagnosis and prognosis. Semin Ophthalmol. 2009, 24, 113-121.
  4. Havens, S.J.; Gulati, V. Neovascular Glaucoma. Dev Ophthalmol. 2016, 55, 196-204.
  5. Barac, I.R.; Pop, M.D.; Gheorghe, A.I.; Taban, C. Neovascular Secondary Glaucoma, Etiology and Pathogenesis. Rom J Ophthalmol. 2015, 59, 24-28.
  6. Tripathi, R.C.; Li, J.; Tripathi, B.J.; Chalam, K.V.; Adamis, A.P. Increased level of vascular endothelial growth factor in aqueous humor of patients with neovascular glaucoma. Ophthalmology. 1998, 105, 232-237.
  7. Simha, A.; Aziz, K.; Braganza, A.; Abraham, L.; Samuel, P.; Lindsley, K.B. Anti-vascular endothelial growth factor for neovascular glaucoma. Cochrane Database Syst Rev. 2020, 2, CD007920.
  8. Sun, Y.; Liang, Y.; Zhou, P.; Wu, H.; Hou, X.; Ren, Z.; Li, X.; Zhao, M. Anti-VEGF treatment is the key strategy for neovascular glaucoma management in the short term. BMC Ophthalmol. 2016, 16, 150.
  9. Weinreb, R.N.; Khaw, P.T. Primary open-angle glaucoma. Lancet. 2004, 363, 1711-1720.
  10. Quigley, H.A.; Broman, A.T. The number of people with glaucoma worldwide in 2010 and 2020. Br J Ophthalmol. 2006, 90, 262-267.
  11. Dietze, J.; Blair, K.; Havens, S.J. Glaucoma. In StatPearls: Treasure Island (FL), 2021.

In discussion, the following references were added (Lines 310).

  1. McDonnell, F.; Dismuke, W.M.; Overby, D.R.; Stamer, W.D. Pharmacological regulation of outflow resistance distal to Schlemm's canal. Am J Physiol Cell Physiol. 2018, 315, C44-C51.
  2. Carreon, T.; van der Merwe, E.; Fellman, R.L.; Johnstone, M.; Bhattacharya, S.K. Aqueous outflow - A continuum from trabecular meshwork to episcleral veins. Prog Retin Eye Res. 2017, 57, 108-133.

Figure 1 legend: please specify the meaning of red and blue arrows.

Response: Thank you for your advice. We have added the following sentences in the end of Figure 1 legend.

Lines 206

A white circle indicates the trabecular meshwork. A blue and a red arrow indicates Schlemm’s canal and the collecting duct, respectively. An orange arrow indicates the trabecular meshwork.

Thank you